# Learning Invariants through Soft Unification

## Abstract

Human reasoning involves recognising common underlying principles across many examples by utilising variables. The by-products of such reasoning are invariants that capture patterns across examples such as "if someone went somewhere then they are there" without mentioning specific people or places. Humans learn what variables are and how to use them at a young age, and the question this paper addresses is whether machines can also learn and use variables solely from examples without requiring human pre-engineering. We propose Unification Networks that incorporate soft unification into neural networks to learn variables and by doing so lift examples into invariants that can then be used to solve a given task. We evaluate our approach on four datasets to demonstrate that learning invariants captures patterns in the data and can improve performance over baselines.

## 1 Introduction

Humans have the ability to process symbolic knowledge and maintain symbolic thought (Unger & Deacon, 1998). When reasoning, humans do not require combinatorial enumeration of examples but instead utilise invariant patterns with placeholders replacing specific entities. Symbolic cognitive models (Lewis, 1999) embrace this perspective with the human mind seen as an information processing system operating on formal symbols such as reading a stream of tokens in natural language. The language of thought hypothesis (Morton & Fodor, 1978) frames human thought as a structural construct with varying sub-components such as "X went to Y". By recognising what varies across examples, humans are capable of lifting examples into invariant principles that account for other instances. This symbolic thought with variables is learned at a young age through symbolic play (Piaget, 2001). For instance a child learns that a sword can be substituted with a stick (Frost et al., 2004) and engage in pretend play.

Although variables are inherent in models of computation and symbolic formalisms, as in first-order logic (Russell & Norvig, 2016), they are pre-engineered and used to solve specific tasks by means of unification or assignments that bound variables to given values. However, when learning from data only, being able to recognise when and which symbols should take on different values, i.e. symbols that can act as variables, is crucial for lifting examples into general principles that are invariant across multiple instances. Figure 1 shows the invariant learned by our approach: if someone is the same thing as someone else then they have the same colour. With this invariant, our approach can solve *all* of the training and test examples in task 16 of the bAbI dataset (Weston et al., 2016).

$X$:bernhard is a $Y$:frog
$Z$:lily is a $Y$:frog
$Z$:lily is $A$:green
___
what colour is $X$:bernhard
___
$A$:green

Figure 1: Invariant learned for bAbI task 16, basic induction, where $X$:bernhard denotes a variable with default symbol bernhard. This single invariant accounts for all the training and test examples.

In this paper we address the question of whether a machine can learn and use the notion of a *variable*, i.e. a symbol that can take on different values. For instance, given an example of the form "bernhard is a frog" the machine would learn that the token "bernhard" could be *someone* else and the token "frog" could be *something* else. If we consider unification a selection of the most appropriate value for a variable given a choice of values, we can reframe it as a form of attention. Attention models (Bahdanau et al., 2015; Luong et al., 2015; Chaudhari et al., 2019) allow neural networks to focus, attend to certain parts of the input often for the purpose of selecting a relevant portion. Since attention mechanisms are also differentiable they are often jointly

learned within a task. This perspective motivates our idea of a unification mechanism that utilises attention and is therefore fully differentiable which we refer to as *soft unification*.

Hence, we propose an end-to-end differentiable neural network approach for learning and utilising the notion of a variable that in return can lift examples into invariants used by the network to perform reasoning tasks. Specifically, we (i) propose a novel architecture capable of learning and using variables by lifting a given example through soft unification, (ii) present the empirical results of our approach on four datasets and (iii) analyse the learned invariants that capture the underlying patterns present in the tasks. Our implementation using Chainer (Tokui et al., 2015) is publicly available at `[link removed](anonymous link provided with submission)`.

## 2 SOFT UNIFICATION

Reasoning with variables involves identifying what variables are, the setting in which they are used as well as the process by which they are assigned values. When the varying components, i.e. variables, of an example are identified, the remaining structure can be lifted into an invariant which then accounts for multiple other instances.

**Definition 1** (Variable). Given a set of symbols $\mathbb{S}$, a variable $X$ is defined as a pair $X \triangleq (\mathrm{x}, s_d)$ where $s_d \in \mathbb{S}$ is the *default symbol* of the variable and x is a discrete random variable of which the support is $\mathbb{S}$. The representation of a variable $\phi_V(X{:}s_d)$ is equal to the expected value of the corresponding random variable x given the default symbol $s_d$:

$$\phi_V(X{:}s_d) = \mathbb{E}_{\mathrm{x} \sim P}[\phi(x)] = \sum_{x \in \mathbb{S}} P(\mathrm{x} = x | s_d)\phi(x) \tag{1}$$

where $\phi : \mathbb{S} \to \mathbb{R}^d$ is a $d$-dimensional real-valued feature of a symbol $s$.

For example, $\phi$ could be an embedding and $\phi_V(X{:}s_d)$ would become a weighted sum of symbol embeddings as in conventional attention models. The default symbol of a variable is intended to capture the variable's bound *meaning* following the idea of referants by Frege (1948). We denote variables using $X$, $Y$, $A$ etc. such as $X$:bernhard where $X$ is the name of the variable and bernhard the default symbol as shown in Figure 1.

**Definition 2** (Invariant). Given a structure (e.g. list, grid) $\mathcal{G}$ over $\mathbb{S}$, an invariant is a pair $I \triangleq (G, \psi)$ where $G \in \mathcal{G}$ is the invariant example such as a tokenised story with tokens as symbols and $\psi : \mathbb{S} \to [0, 1]$ is a function representing the degree to which the symbol is considered a variable. Thus, the final representation of a symbol $s$ included in $G$, $\phi_I(s)$ is:

$$\phi_I(s) = (1 - \psi(s))\phi(s) + \psi(s)\phi_V(X{:}s) \tag{2}$$

the linear interpolation between its representation $\phi(s)$ and its variable bound value with itself as the default symbol $\phi_V(X{:}s)$.

We adhere to the term invariant and refrain from mentioning rules, unground rules, etc. used in logic-based formalisms, e.g. Muggleton & de Raedt (1994), since neither the invariant structure needs to be rule-like nor the variables carry logical semantics. This distinction is clarified in Section 6.

**Definition 3** (Unification). Given an invariant $I$ and an example $K \in \mathcal{G}$, unification binds the variables in $I$ to symbols in $K$. Defined as a function $g : \mathcal{I} \times \mathcal{G} \to \mathcal{G}$, unification binds variables by computing the probability mass functions, $P$ in equation 1, and returns the unified representation using equation 2. The probability mass function of a variable $X{:}s_d$ is:

$$P(\mathrm{x} = x | s_d) = \mathrm{softmax}(\phi_U(s_d)\phi_U(K)^T), \ K = \text{support of x} \tag{3}$$

where $\phi_U : \mathbb{S} \to \mathbb{R}^d$ is the unifying feature of a symbol and $\phi_U(K) \in \mathbb{R}^{|K| \times d}$ is applied element wise to symbols in $K$. If $g$ is differentiable, it is referred to as soft unification.

We distinguish $\phi$ from $\phi_U$ to emphasise that the unifying properties of the symbols might be different from their representations. For example, $\phi(\text{bernhard})$ could represent a specific person whereas $\phi_U(\text{bernhard})$ the notion of *someone*.

Overall soft unification incorporates 3 learnable components: $\phi, \psi, \phi_U$ which denote the base features, variableness and unifying features of a symbol respectively. Given an upstream, potentially

task specific, network $f : \mathcal{G} \to \mathbb{S}$, an invariant $I \in \mathcal{I}$ and an input example $K \in \mathcal{G}$ with a corresponding desired output $a \in \mathbb{S}$, the following holds:

$$f \circ g(I, K) = f(K) = a \qquad (4)$$

where $f$ now predicts based on the unified representation produced by $g$. In this work, we focus on $g$, the invariants it produces together with the interaction of $f \circ g$.

## 3 UNIFICATION NETWORKS

Since soft unification is end-to-end differentiable, it can be incorporated into existing task-specific upstream architectures. We present 3 architectures that model $f \circ g$ using multi-layer perceptrons (MLP), convolutional neural networks (CNN) and memory networks (Weston et al., 2015) to demonstrate the flexibility of our approach. In all cases, the $d$ dimensional representation of symbols are learnable embeddings $\phi(s) = O[s]^T \boldsymbol{E}$ with $\boldsymbol{E} \in \mathbb{R}^{|\mathbb{S}| \times d}$ randomly initialised by $\mathcal{N}(0, 1)$ and $O[s]$ the one-hot encoding of the symbol. The variableness of symbols is a learnable weight $\psi_w(s) = \sigma(w_s)$ where $\boldsymbol{w} \in \mathbb{R}^{|\mathbb{S}|}$ and $\sigma$ is the sigmoid function. We consider every symbol independently a variable irrespective of its surrounding context and leave further contextualised formulations as future work. The underlying intuition of this configuration is that a *useful* symbol for a correct prediction might need to take on other values for different inputs. This usefulness can be viewed as the inbound gradient to the corresponding $w_s$ parameter and $\psi_w(s)$ acting as a gate. For further model details including the size of the embeddings, please refer to Appendix A.

**Unification MLP (UMLP)** ($f$: MLP, $g$: RNN) We combine soft unification into a multi-layer perceptron to process fixed length inputs. In this case, the structure $\mathcal{G}$ is a sequence of symbols with a fixed length $l$, e.g. a sequence of digits 4234. Given an embedded input $\phi(\boldsymbol{k}) \in R^{l \times d}$, the upstream MLP computes the output symbol based on the flattened representations $f(\phi(\boldsymbol{k})) = \text{softmax}(\boldsymbol{h}\boldsymbol{E}^T)$ where $\boldsymbol{h} \in \mathbb{R}^d$ is the output of the last layer. However, to compute the unifying features $\phi_U$, definition 3, $g$ uses a bi-directional GRU (Cho et al., 2014) running over $\phi(\boldsymbol{k})$ such that $\phi_U(k) = \boldsymbol{W}_U \Phi(k)$ where $\Phi(k) \in \mathbb{R}^d$ is the hidden state of the GRU at symbol $k$ and $\boldsymbol{W}_U \in \mathbb{R}^{d \times d}$ is a learnable parameter. This model emphasises the flexibility around the boundary of $f \circ g$ and that the unifying features can be computed in any differentiable manner.

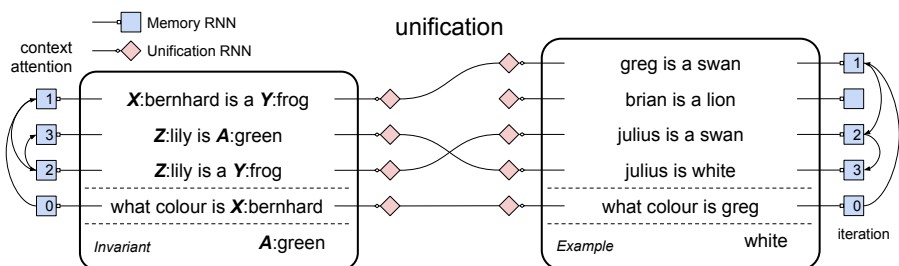

Figure 2: Graphical overview of soft unification within a memory network. Each sentence is processed by two bi-directional RNNs for memory and unification. At each iteration the context attention selects which sentences to unify and the invariant produces the same answer as the example.

**Unification CNN (UCNN)** ($f$: CNN, $g$: CNN) Given a grid of embedded symbols $\phi(\boldsymbol{K}) \in \mathbb{R}^{w \times h \times d}$ where $w$ is the width and $h$ the height, we use a convolutional neural network such that the final prediction is $f(\phi(\boldsymbol{K})) = \text{softmax}((\boldsymbol{W}\boldsymbol{h} + \boldsymbol{b})\boldsymbol{E}^T)$ where $\boldsymbol{h}$ this time is the result of global max pooling and $\boldsymbol{W}, \boldsymbol{b}$ are learnable parameters. We also model $g$ using a *separate* convolutional network with the same architecture as $f$ and set $\phi_U(k) = c_2(\text{relu}(c_1(k)))$ where $c_1, c_2$ are the convolutional layers. The grid is padded with 0s to obtain $w \times h \times d$ after each convolution such that every symbol has a unifying feature. This model conveys how soft unification can be adapted to the specifics of the domain for example by using a convolution in a spatially structured input.

**Unification Memory Networks (UMN)** ($f$: MemNN, $g$: RNN) Soft unification does not need to happen prior to $f$ in a $f \circ g$ fashion but can also be incorporated at any intermediate stage multiple times. To demonstrate this ability, we unify the symbols at different memory locations at each iteration of a Memory Network (Weston et al., 2015). Memory networks can handle a list of lists

Table 1: Sample context, query and answer triples and their training sizes *per task*. For distribution of generated number of examples per task on Sequence and Grid data refer to Appendix B.

| Dataset | Context | Query | Answer | Training Size |
|---------|---------|-------|--------|---------------|
| Sequence | 8384 | duplicate | 8 | $\leq$ 1k, $\leq$ 50 |
| Grid | 0 0 3
0 1 6
8 5 7 | corner | 7 | $\leq$ 1k, $\leq$ 50 |
| bAbI | Mary went to the kitchen.
Sandra journeyed to the garden. | Where is Mary? | kitchen | 1k, 50 |
| Logic | p(X) ← q(X).
q(a). | p(a). | True | 1k, 10k, 50 |

structure such as a tokenised story as shown in Figure 2. The memory network $f$ uses the final hidden state of a bi-directional GRU (outer squares in Figure 2) as the sentence representations to compute a context attention. At each iteration, we unify the words between the attended sentences using the same approach in UMLP with another bi-directional GRU (inner diamonds in Figure 2) for unifying features $\phi_U(\text{bernhard}) = \boldsymbol{W}_U \Phi(\text{bernhard})$. Following equation 2, the new unified representation of the memory slot is computed and $f$ uses it to perform the next iteration. Concretely, $g$ produces an unification tensor $\mathbf{U} \in \mathbb{R}^{M \times m \times N \times d}$ where $M$ and $m$ is the number of sentences and words in the invariant respectively, and $N$ is the number of sentences in the example such that after the context attentions are applied over $M$ and $N$, we obtain $\phi(\boldsymbol{k}) \in \mathbb{R}^{m \times d}$ as the unified sentence at that iteration. Note that unlike in the UMLP case, the sentences can be of varying length. The prediction is then $\text{softmax}(\boldsymbol{W}\boldsymbol{h}_I^J + \boldsymbol{b})$ where $\boldsymbol{h}_I^J$ is the hidden state of the invariant after $J$ iterations. This setup, however, requires pre-training $f$ such that the context attentions match the correct sentences.

A task might contain different questions such as "Where is X?" and "Why did X go to Y?". To let the models differentiate between questions and potentially learn different invariants, we extend them with a repository of invariants $I \in \mathbb{I}$ and aggregate the predictions from each invariant. One simple approach is to sum the predictions of the invariants $\sum_{I \in \mathbb{I}} f \circ g(I, K)$ used in UMLP and UCNN. Another approach could be to use features from the invariants such as memory representations in the case of UMN. For UMN, we weigh the predictions using a bilinear attention $\eta$ based on the hidden states at the first iteration $\boldsymbol{h}_I^0$ and $\boldsymbol{h}_K^0$ such that $\eta = \text{softmax}(\boldsymbol{h}_I^0 \boldsymbol{W} \boldsymbol{h}_K^{0^T})$. To initially form the repository of invariants, we use the bag-of-words representation of the questions and find the most dissimilar ones based on their cosine similarity as a heuristic to obtain varied examples.

## 4 DATASETS

We use 4 datasets consisting of context, query and an answer $(\boldsymbol{C}, \boldsymbol{q}, a)$: fixed length sequences of symbols, shapes of symbols in a grid, story based natural language reasoning with the bAbI (Weston et al., 2016) dataset and logical reasoning represented as logic programs, examples shown in Table 1 with further samples in Appendix B. In each case we use an appropriate model: UMLP for fixed length sequences, UCNN for grid and UMN for iterative reasoning. We use synthetic datasets of which the data generating distributions are known to evaluate not only the quantitative performance but also the quality of the invariants learned by our approach.

**Fixed Length Sequences** We generate sequences of length $l = 4$ with 8 unique symbols represented as digits to predict (i) a constant, (ii) the head of the sequence, (iii) the tail and (iv) the duplicate symbol. We randomly generate 1000 triples and then only take the unique ones to ensure the test split contains unseen examples. The training is then performed over a 5-fold cross-validation.

**Grid** To spatially organise symbols, we generate a grid of size $3 \times 3$ with 8 unique symbols organised into $2 \times 2$ box of identical symbol, a vertical, diagonal or horizontal sequence of length 3, a cross or a plus shape and a triangle. In each task we predict (i) the identical symbol, (ii) the head of the sequence, (iii) the centre of the cross or plus and (iv) the corner of the triangle respectively. We follow the same procedure from sequences and randomly generate 1000 discarding duplicate triples.

**bAbI** The bAbI dataset has become a standard benchmark for evaluating memory based networks. It consists of 20 synthetically generated natural language reasoning tasks (refer to Weston et al.

(2016) for task details). We take the 1k English set and use 0.1 of the training set as validation. Each token is lower cased and considered a unique symbol. Following previous works (Seo et al., 2017; Sukhbaatar et al., 2015), we take multiple word answers also to be a unique symbol in $\mathbb{S}$.

**Logical Reasoning** To demonstrate the flexibility of our approach and distinguish our notion of a variable from that used in logic based formalisms, we generate logical reasoning tasks in the form of logic programs using the procedure by Cingillioglu & Russo (2019). The tasks involve learning $f(C, Q) = \text{True} \leftrightarrow C \vdash Q$ over 12 classes of logic programs exhibiting varying paradigms of logical reasoning including negation by failure (Clark, 1978). We generate 1k and 10k logic programs per task for training with 0.1 as validation and another 1k for testing. We set the arity of literals to 1 or 2 using one random character from the English alphabet for predicates *and* constants, e.g. $p(p)$ and an upper case character for logical variables, e.g. $p(\mathtt{X})$.

## 5 EXPERIMENTS

We probe three aspects of soft unification: the impact of unification on performance over unseen data, the effect of multiple invariants and data efficiency. To that end, we train UMLP and UCNN with and without unification, UMN with pre-training using 1 or 3 invariants over either the entire training set or only 50 examples. Every model is trained 3 times via back-propagation using Adam (Kingma & Ba, 2015) on an Intel Core i7-6700 CPU using the following objective function:

$$J = \lambda_K \mathcal{L}_{\text{nll}}(f(K), a) + \lambda_U \left[ \mathcal{L}_{\text{nll}}(f \circ g(I, K), a) + \tau \sum_{s \in \mathbb{S}} \psi_w(s) \right] \quad (5)$$

where $\mathcal{L}_{\text{nll}}$ is the negative log-likelihood with sparsity regularisation over $\psi$ at $\tau = 0.1$ to discourage the models from utilising spurious number of variables. For UMLP and UCNN, we set $\lambda_K = 0$, $\lambda_U = 1$ for training just the unified output and the converse for the non-unifying versions. To pre-train the UMN, we start with $\lambda_K = 1$, $\lambda_U = 0$ for 40 epochs then set $\lambda_U = 1$ to jointly train the unified output. For iterative tasks, the mean squared error between hidden states $(h_I^j - h_K^j)^2$ at each iteration $j$ and, in the strongly supervised cases, the negative log-likelihood for the context attentions using provided supporting facts are also added to the objective function. Further details such as batch size and total number of epochs are available in Appendix C.

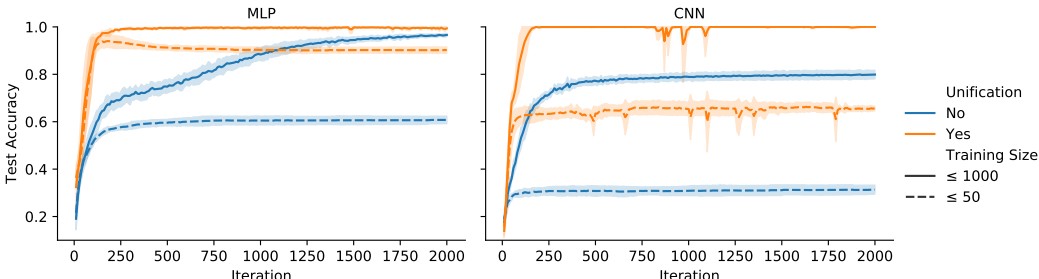

Figure 3: Test accuracy over iterations for Unification MLP and Unification CNN models with 1 invariant versus no unification. We observe that with soft unification the models achieve higher accuracy with fewer iterations than their plain counterparts on both per task training sizes.

Figure 3 portrays how soft unification generalises better to unseen examples in test sets - the same sequence or grid never appears in both the training and test sets as outlined in Section 4 - over plain models. Despite $f \circ g$ having more trainable parameters than $f$ alone, the models with unification not only maintain higher accuracy in each iteration and solve the tasks in as few as 250 iterations with $\leq 1000$ training examples but also improve accuracy by $\approx 0.3$ when trained with only $\leq 50$ per task. We believe soft unification architecturally biases the models towards learning structural patterns which in return achieves better results on recognising common patterns of symbols across examples. Results with multiple invariants are identical and the models seem to ignore the extra invariants due to the fact that the tasks can be solved with a single invariant and the regularisation applied on $\psi$ zeroing out unnecessary invariants; further results in Appendix D. The fluctuations in accuracy around iterations 750 to 1000 in UCNN are also caused by penalising $\psi$ which forces the model to relearn the task with less variables half way through training.

Table 2: Aggregate error rates (%) on bAbI 1k for UMN and N2N, GN2N, MemNN by Sukhbaatar et al. (2015), Liu & Perez (2017) and Weston et al. (2015) respectively. Full comparison on individual tasks are available in Appendix D.

| Training Size | 1k | | | | 50 | 1k | | | | |
| --- | --- | --- | --- | --- | --- | --- | --- | --- | --- | --- |
| Supervision | Weak | | Strong | | | Weak | | | | Strong |
| # Invs / Model | 1 | 3 | 1 | 3 | 3 | N2N | GN2N | EntNet | QRN | MemNN |
| Mean | 19.1 | 20.5 | 6.3 | **6.0** | 27.6 | 13.9 | 12.7 | 29.6 | 11.3 | 6.7 |
| # > 5% | 8 | 10 | 4 | 4 | 17 | 11 | 10 | 15 | 5 | 4 |

Following Tables 2 and 3, we observe a trend of better performance through strong supervision, more data per task and using only 1 invariant. We believe strong supervision aids with selecting the correct sentences to unify and in a weak setting the model attempts to unify arbitrary context sentences often failing to follow the iterative reasoning chain. The increase in performance with more data and strong supervision is consistent with previous work reflecting how $f \circ g$ can be bounded by the efficacy of $f$ modelled as a memory network. As a result, only in the supervised case do we observe a minor improvement over MemNN by 0.7 in Table 2 and no improvement in the weak case over comparable memory based networks in Table 3. This dependency on $f$ also limits the ability of $f \circ g$ to learn from 50 examples per task failing 17/20 and 12/12 of bAbI and logical reasoning tasks respectively. The increase in error rate with 3 invariants, we speculate, stems from having more parameters and more pathways in the model rendering training more difficult.

Table 3: Aggregate task error rates (%) on the logical reasoning dataset for UMN and DMN, IMA by Cingillioglu & Russo (2019). Strong supervision, more data and only 1 invariant seem to improve the performance of UMN over plain iterative models. Individual task results are in Appendix D.

| Training Size | 1k | | | | 10k | | | | | 50 | 20k | |
| --- | --- | --- | --- | --- | --- | --- | --- | --- | --- | --- | --- | --- |
| Supervision | Weak | | Strong | | Weak | | Strong | | | | Weak | |
| Arity | 1 | 2 | 1 | 2 | 1 | 2 | 1 | 2 | 2 | 2 | 2 | 2 |
| # Invs / Model | 1 | 3 | 1 | 3 | 1 | 3 | 1 | 1 | 3 | 3 | DMN | IMA |
| Mean | 36.4 | 39.3 | 14.3 | 28.9 | 21.5 | 31.8 | **2.4** | 12.2 | 16.0 | 47.1 | 21.2 | 9.1 |
| # > 5% | 9 | 11 | 7 | 11 | 7 | 10 | 1 | 5 | 9 | 12 | 11 | 5 |

## 6 ANALYSIS

After training, we can extract the learned invariants by applying a threshold on $\forall s \in \mathbb{S} : \psi(s) > t$ indicating whether a symbol is used as a variable or not. We set $t = 0.0$ for all datasets except for bAbI, we use $t = 0.1$. The magnitude of this threshold seems to depend on the amount of regularisation $\tau$, equation 5, and the number of training steps along with batch size all controlling how much $\psi$ is pushed towards 0. Sample invariants shown in Figure 4 describe the common patterns present in the tasks with parts that contribute towards the final answer becoming variables. Extra symbols such as `is` or `travelled` do not emerge as variables, as shown in Figure 4a; we attribute this behaviour to the fact that changing the token `travelled` to `went` does not influence the prediction but changing the action, the value of **Z**:left to 'picked' does. However, based on random initialisation, our approach can convert an arbitrary symbol into a variable and let $f$ compensate for the unifications it produces. For example, the invariant "**X**:8 5 2 2" could predict the tail of another example by unifying the head with the tail using $\phi_U$, equation 3, of those symbols. Further examples are shown in Appendix D. Pre-training $f$ as done in UMN seems to produce more robust and consistent invariants compared to immediately training $f \circ g$ since, we speculate, by equation 4 $f$ might encourage $g(I, K) \approx K$.

**Interpretability versus Ability** A desired property of interpretable models is transparency (Lipton, 2018). A novel outcome of the learned invariants in our approach is that they provide an approximation of the underlying general principle present in the *data* such as the structure of multi-hop reasoning shown in Figure 4e. However, certain aspects regarding the ability of the model such as how it performs temporal reasoning, are still hidden inside $f$. In Figure 4b, although we observe **Z**:morning as a variable, the overall learned invariant captures nothing about how changing the value of **Z**:morning alters the behaviour of $f$. The model might look *before* or *after* a certain time point

|  |  |
|---|---|
| *Y*:john *Z*:left the *X*:football | this *Z*:morning *X*:bill went to the *Y*:school |
| *Y*:john travelled to the *A*:office | yesterday *X*:bill journeyed to the *A*:park |
| where is the *X*:football | where was *X*:bill before the *Y*:school |
| *A*:office | *A*:park |

(a) bAbI task 2, two supporting facts. The model also learns *Z*:left since people can also drop or pick up objects potentially affecting the answer.

(b) bAbI task 14, time reasoning. *X*:bill and *Y*:school are recognised as variables alongside *Z*:morning capturing *when* someone went which is crucial to this task.

|  |  |  |  |
|---|---|---|---|
| 5 8 6 4 const | 2 |
| *X*:8 3 3 1 head | *X*:8 |
| 8 3 1 *Y*:5 tail | *Y*:5 |
| *Z*:1 4 3 *Z*:1 dup | *Z*:1 |

| | | |
|---|---|---|
| 0 *X X* | 0 1 0 | 0 0 1 |
| 0 *X X* | 6 *Y* 8 | 0 5 4 |
| 0 0 0 | 0 7 0 | 7 8 *X* |
| box | centre | corner |

$$X{:}\mathrm{i}\,(\,\mathrm{T}\,)\leftarrow Z{:}\mathrm{l}\,(\,\mathrm{T}\,),$$
$$Z{:}\mathrm{l}\,(\,\mathrm{U}\,)\leftarrow R{:}\mathrm{x}\,(\,\mathrm{U}\,),$$
$$R{:}\mathrm{x}\,(\,\mathrm{K}\,)\leftarrow S{:}\mathrm{n}\,(\,\mathrm{K}\,),$$
$$S{:}\mathrm{n}\,(\,Y{:}\mathrm{o}\,)\vdash X{:}\mathrm{i}\,(\,Y{:}\mathrm{o}\,)$$

(c) Successful invariants learned with UMLP using 50 training examples only shown as $(C, Q, a)$.

(d) Successful invariants learned with UCNN. Variable default symbols are omitted for clarity.

(e) Logical reasoning task 5 with arity 1. The model captures how *S*:n could entail *X*:i in a chain.

Figure 4: Invariants learned across the four datasets using the three architectures. For iterative reasoning datasets, bAbI and logical reasoning, they are taken from strongly supervised UMN.

*X*:bill went somewhere depending what *Z*:morning binds to. Without the regularising term on $\psi(s)$, we initially noticed the models using, one might call extra, symbols as variables and binding them to the same value occasionally producing unifications such as "bathroom bathroom to the bathroom" and still $f$ predicting, perhaps unsurprisingly, the correct answer as bathroom. Hence, regularising $\psi$ with the correct amount $\tau$ in equation 5 to reduce the capacity of unification seems critical in extracting not just any invariant but one that represents the common structure.

Soft unification from equation 3 reveals three main patterns: one-to-one, one-to-many or many-to-one bindings as shown in Figure 5; further examples are in Appendix D. Figure 5a captures what one might expect unification to look like where variables unify with their corresponding counterparts, e.g. *X*:bernhard with brian and *Y*:frog with lion. However, occasionally the model can optimise to use less variables and *squeeze* the required information into a single variable, for example by binding *Y*:bathroom to john and kitchen as shown in Figure 5b. We believe this occurs due to the sparsity constraint on $\psi(s)$ encouraging the model to be as conservative as possible. Finally, if there are more variables than needed as in Figure 5c, we observe a many-to-one binding with *Y*:w and *Z*:e mapping to the same constant $q$. This behaviour begs the question how does the model differentiate between $p(q)$ and $p(q, q)$. We speculate the model uses the magnitude of $\psi(w) = 0.037$ and $\psi(e) = 0.042$ to encode the difference despite both variables unifying with the same constant.

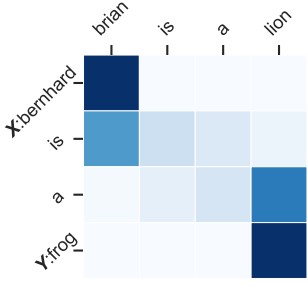

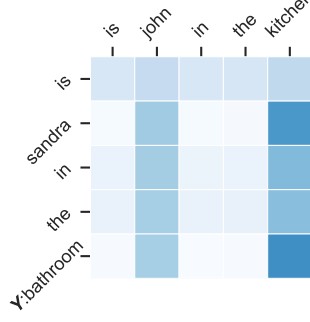

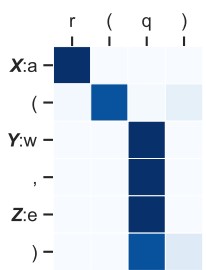

(a) bAbI task 16. A one-to-one mapping is created between variables *X*:bernhard with brian and *Y*:frog with lion.

(b) bAbI task 6. Only *Y*:bathroom is recognised as variable creating a one-to-many binding to capture the same information.

(c) Logical reasoning task 1. An arity 2 predicate is forced to bind with arity 1 creating a many-to-one binding.

Figure 5: Variable bindings produced from equation 3. Darker cells indicate higher attention values.

# 7 RELATED WORK

Learning an underlying general principle in the form of an invariant is often the means for arguing generalisation in neural networks. For example, Neural Turing Machines (Graves et al., 2014) are tested on previously unseen sequences to support the view that the model might have captured the underlying pattern or algorithm. In fact, Weston et al. (2015) claim "MemNNs can discover simple linguistic patterns based on verbal forms such as (X, dropped, Y), (X, took, Y) or (X, journeyed to, Y) and can successfully generalise the meaning of their instantiations." However, this claim is based on the output of $f$ and unfortunately it is unknown whether the model has truly learned such a representation or indeed is utilising it. Our approach sheds light to this ambiguity and presents these linguistic patterns explicitly as invariants ensuring their utility through $g$ without solely analysing the output of $f$ on previously unseen symbols. Although we associate these invariants with our existing understanding of the task to mistakenly anthropomorphise the machine, for example by thinking it has learned $X$:mary as *someone*, it is important to acknowledge that these are just symbolic patterns. In these cases, our interpretations do not necessarily correspond to any understanding of the machine, relating to the Chinese room argument made by Searle (1980).

Learning invariants by lifting ground examples is related to least common generalisation (Reynolds, 1970) by which inductive inference is performed on facts (Shapiro, 1981) such as generalising *went(mary,kitchen)* and *went(john,garden)* to *went(X,Y)*. Unlike in a predicate logic setting, our approach allows for soft alignment and therefore generalisation between varying length sequences. Existing neuro-symbolic systems (Broda et al., 2002) focus on inducing rules that adhere to *given* logical semantics of what variables and rules are. For example, $\delta ILP$ (Evans & Grefenstette, 2018) constructs a network by rigidly following the given semantics of first-order logic. Similarly, Lifted Relational Neural Networks (Sourek et al., 2015) ground first-order logic rules into a neural network while Neural Theorem Provers (Rocktschel & Riedel, 2017) build neural networks using backward-chaining (Russell & Norvig, 2016) on a given background knowledge base with templates. However, the notion of a variable is pre-defined rather than learned with a focus on presenting a practical approach to solving certain problems, whereas our motivation stems from a cognitive perspective.

At first it may seem the learned invariants, Section 6, make the model more interpretable; however, this transparency is not of the model $f$ but of the data. The invariant captures patterns that potentially approximates the data generating distribution but we still do not know *how* the model $f$ uses them upstream. Thus, from the perspective of explainable artificial intelligence (XAI) (Adadi & Berrada, 2018), learning invariants or interpreting them do not constitute an explanation of the reasoning model $f$ even though "if *someone* goes *somewhere* then they are there" might look like one. Instead, it can be perceived as causal attribution (Miller, 2019) in which someone being somewhere is attributed to them going there. This perspective also relates to gradient based model explanation methods such as Layer-Wise Relevance Propagation (Bach et al., 2015) and Grad-CAM (Selvaraju et al., 2017; Chattopadhay et al., 2018). Consequently, a possible view on $\psi$, equation 2, is a gradient based usefulness measure such that a symbol utilised upstream by $f$ to determine the answer becomes a variable similar to how a group of pixels in an image contribute more to its classification.

Finally, one can argue that our model maintains a form of counterfactual thinking (Roese, 1997) in which soft unification $g$ creates counterfactuals on the invariant example to alter the output of $f$ towards the desired answer, equation 4. The question where Mary would have been if Mary had gone to the garden instead of the kitchen is the process by which an invariant is learned through multiple examples during training. This view relates to methods of causal inference (Pearl, 2019; Holland, 1986) in which counterfactuals are vital as demonstrated in structured models by Pearl (1999).

# 8 CONCLUSION

We presented a new approach for learning variables and lifting examples into invariants through the usage of soft unification. Evaluating on four datasets, we analysed how Unification Networks perform comparatively to existing similar architectures while having the benefit of lifting examples into invariants that capture underlying patterns present in the tasks. Since our approach is end-to-end differentiable, we plan to apply this technique to multi-modal tasks in order to yield multi-modal invariants for example in visual question answering.

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

# A    MODEL DETAILS

## A.1    UNIFICATION MLP & CNN

**Unification MLP (UMLP)** To model $f$ as a multi-layer perceptron, we take symbol embeddings of size $d = 16$ and flatten sequences of length $l = 4$ into an input vector of size $\phi(\boldsymbol{k}) \in \mathbb{R}^{64}$. The MLP consists of 2 hidden layers with $\tanh$ non-linearity of sizes $2d$ and $d$ respectively. To process the query, we concatenate the one-hot encoding of the task id to $\phi(\boldsymbol{k})$ yielding a final input of size $64 + 4 = 68$. For unification features $\phi_U$, we use a bi-directional GRU with hidden size $d$ and an initial state of $\mathbf{0}$. The hidden state at each symbol is taken with a linear transformation to give $\phi_U(s) = \boldsymbol{W}_U \Phi(s)$ where $\Phi(s)$ is the hidden state of the biGRU. The variable assignment is then computed as an attention over the according to equation 3.

**Unification CNN (UCNN)** We take symbols embeddings of size $d = 32$ to obtain an input grid $\phi(\boldsymbol{K}) \in \mathbb{R}^{3 \times 3 \times 32}$. Similar to UMLP, for each symbol we append the task id as a one-hot vector to get an input of shape $3 \times 3 \times (32 + 4)$. Then $f$ consists of 2 convolutional layers with $d$ filters each, kernel size of 3 and stride 1. We use $\mathrm{relu}$ non-linearity in between the layers. We pad the grid with 2 columns and 2 rows to a $5 \times 5$ such that the output of the convolutions yield again a hidden output $\mathbf{H} \in \mathbb{R}^{3 \times 3 \times d}$ of the same shape. As the final hidden output $h$, we take a global max pool to over $\mathbf{H}$ to obtain $\boldsymbol{h} \in \mathbb{R}^d$. Unification function $g$ is modelled identical to $f$ without the max pooling such that $\phi_U(\boldsymbol{K}_{ij}) = \mathbf{H}'_{ij}$ where $\mathbf{H}'$ is the hidden output of the convolutional layers.

## A.2    UNIFICATION MEMORY NETWORKS

Unlike previous architectures, with UMN we interleave $g$ into $f$. We use embedding sizes of $d = 32$ and model $f$ with an iterative memory network. We take the final hidden state of a bi-directional GRU, with initial state $\mathbf{0}$, $\Phi_M$ to represent the sentences of the context $\boldsymbol{C}$ and query $\boldsymbol{q}$ in a $d$-dimensional vector $\boldsymbol{M}_i = \Phi_M(\boldsymbol{C}_i)$ and the query $\boldsymbol{m}_q = \Phi_M(\boldsymbol{q})$. The initial state of the memory network is $\boldsymbol{h}^0 = \boldsymbol{m}_q$. At each iteration $j$:

$$\boldsymbol{A}_i^j = \tanh(\boldsymbol{W}\rho(\boldsymbol{M}_i, \boldsymbol{h}^j) + \boldsymbol{b}) \tag{6}$$

$$\beta^j = \mathrm{softmax}(\boldsymbol{W}\Phi_A(\boldsymbol{A}^j) + \boldsymbol{b}) \tag{7}$$

where $\Phi_A$ is another $d$-dimensional bi-directional GRU and $\rho(\boldsymbol{x}, \boldsymbol{y}) = [\boldsymbol{x}; \boldsymbol{y}; \boldsymbol{x} \odot \boldsymbol{y}; (\boldsymbol{x} - \boldsymbol{y})^2]$ with $\odot$ the element-wise multiplication and $[;]$ the concatenation of vectors. Taking $\beta^j$ as the context attention, we obtain the next state of the memory network:

$$\boldsymbol{h}^{j+1} = \sum_i \beta_i^j \tanh(\boldsymbol{W}\rho(\boldsymbol{M}_i, \boldsymbol{h}^j) + \boldsymbol{b}) \tag{8}$$

and iterate $J$ many times in advance. The final prediction becomes $f(\boldsymbol{C}, \boldsymbol{q}) = \mathrm{softmax}(\boldsymbol{W}\boldsymbol{h}^J + \boldsymbol{b})$. All weight matrices $\boldsymbol{W}$ and bias vectors $\boldsymbol{b}$ are independent but are tied across iterations.

## B GENERATED DATASET SAMPLES

Table 4: Sample context, query and answer triples from sequences and grid tasks.

| Dataset | Task | Context | Query | Answer |
|---------|------|---------|-------|--------|
| Sequence | i | 1488 | constant | 2 |
| Sequence | ii | 6157 | head | 6 |
| Sequence | iii | 1837 | tail | 7 |
| Sequence | iv | 3563 | duplicate | 3 |
| Grid | i | 0 0 0
0 2 2
8 2 2 | box | 2 |
| Grid | ii | 4 0 0
0 7 0
8 0 1 | head | 4 |
| Grid | iii | 0 6 0
1 7 2
0 3 0 | centre | 7 |
| Grid | iv | 8 0 0
5 6 0
2 4 1 | corner | 2 |

Table 5: Training sizes for randomly generated fixed length sequences and grid tasks with 8 unique symbols. The reason for Grid task (i) to be smaller is because there are at most 32 combinations of $2 \times 2$ boxes in a $3 \times 3$ grid with 8 unique symbols.

| Task | Sequences | Grid |
|------|-----------|------|
| i | $704.7 \pm 12.8$ | $25.6 \pm 1.8$ |
| ii | $709.4 \pm 13.8$ | $623.7 \pm 14.1$ |
| iii | $709.7 \pm 14.0$ | $768.2 \pm 12.5$ |
| iv | $624.8 \pm 12.4$ | $795.2 \pm 10.3$ |

## C  TRAINING DETAILS

### C.1  UNIFICATION MLP & CNN

Both unification models are trained on a 5-fold cross-validation over the generated datasets for 2000 iterations with a batch size of 64. We don't use any weight decay and save the training and test accuracies every 10 iterations, as presented in Figure 3.

### C.2  UNIFICATION MEMORY NETWORKS

We again use a batch size of 64 and pre-train $f$ for 40 epochs then $f$ together with $g$ for 260 epochs. We use epochs for UMN since the dataset sizes are fixed. To learn $g$ alongside $f$, we combine error signals from the unification of the invariant and the example. Following equation 4, the objective function not only incorporates the negative log-likelihood $\mathcal{L}_{\mathrm{nll}}$ of the answer but also the mean squared error between intermediate states $\boldsymbol{h}_I^j$ and $\boldsymbol{h}_K^j$ at each iteration as an auxiliary loss:

$$J = \mathcal{L}_{\mathrm{nll}}(f(K), a) + \lambda_U \left[ \mathcal{L}_{\mathrm{nll}}(f \circ g(I, K), a) + \frac{1}{J} \sum_{j=1}^{J} (\boldsymbol{h}_I^j - \boldsymbol{h}_K^j)^2 + \tau \sum_{s \in \mathcal{S}} \psi_w(s) \right] \quad (9)$$

We pre-train by setting $\lambda_U = 0$ for 40 epochs and then set $\lambda_U = 1$. For strong supervision we also compute the negative log-likelihood $\mathcal{L}_{\mathrm{nll}}$ for the context attention $\beta^j$, described in Appendix A, at each iteration using the supporting facts of the tasks. We apply a dropout of 0.1 for all recurrent neural networks used and only for the bAbI dataset weight decay with 0.001 as the coefficient.

# D  FURTHER RESULTS

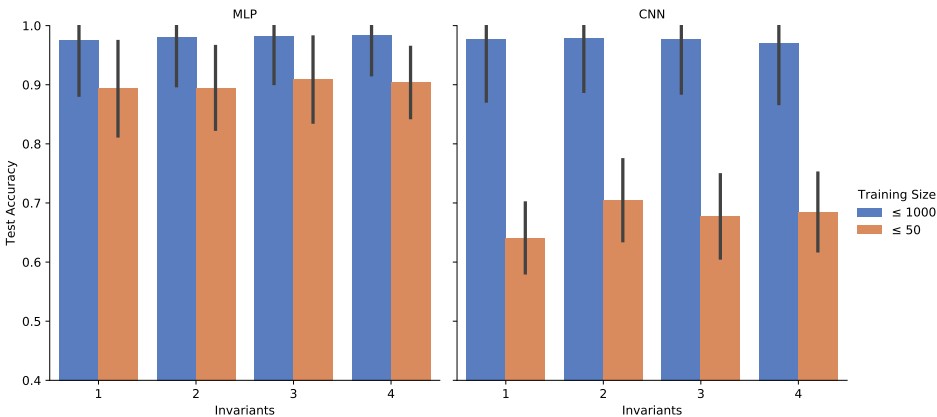

Figure 6: Results of Unification MLP and CNN on increasing number of invariants. There is no impact on performance when more invariants per task are given. Upon closer inspection, we noticed the models ignore the extra invariants and only use 1. We speculate the regularisation $\psi$ encourages the models to use a single 1 invariant.

Table 6: Individual task error rates on bAbI tasks for Unification Memory Networks.

| Supervision | Weak | | Strong | | |
|---|---|---|---|---|---|
| # Invs | 1 | 3 | 1 | 3 | 3 |
| Training Size | 1k | 1k | 1k | 1k | 50 |
| 1 | 0.0 | 0.0 | 0.0 | 0.0 | 1.4 |
| 2 | 65.6 | 63.1 | 0.3 | 0.7 | 30.0 |
| 3 | 67.1 | 62.6 | 1.0 | 2.4 | 39.8 |
| 4 | 0.0 | 0.0 | 0.0 | 0.0 | 37.0 |
| 5 | 3.4 | 4.0 | 0.8 | 1.1 | 26.5 |
| 6 | 0.2 | 0.6 | 0.0 | 0.0 | 18.4 |
| 7 | 22.0 | 22.8 | 10.7 | 11.3 | 22.8 |
| 8 | 10.3 | 8.5 | 7.4 | 7.6 | 24.7 |
| 9 | 0.1 | 25.7 | 0.0 | 0.0 | 33.8 |
| 10 | 0.1 | 2.0 | 0.0 | 0.3 | 32.6 |
| 11 | 0.0 | 0.0 | 0.0 | 0.0 | 11.9 |
| 12 | 0.0 | 0.1 | 0.0 | 0.0 | 21.3 |
| 13 | 2.1 | 3.7 | 0.0 | 0.1 | 5.8 |
| 14 | 19.7 | 13.5 | 0.5 | 0.1 | 54.8 |
| 15 | 0.0 | 0.7 | 0.0 | 0.0 | 0.0 |
| 16 | 55.2 | 56.2 | 0.0 | 0.0 | 39.7 |
| 17 | 39.2 | 49.0 | 51.1 | 49.3 | 48.8 |
| 18 | 4.4 | 8.0 | 0.6 | 0.5 | 10.4 |
| 19 | 91.8 | 89.6 | 53.9 | 46.7 | 90.2 |
| 20 | 0.0 | 0.0 | 0.0 | 0.0 | 2.7 |
| Mean | 19.1 | 20.5 | 6.3 | **6.0** | 27.6 |
| Std | 27.9 | 27.0 | 15.6 | 14.3 | 21.0 |
| # > 5% | 8 | 10 | 4 | 4 | 17 |

Table 7: Comparison of individual task error rates (%) on the bAbI (Weston et al., 2016) dataset of the best run. We preferred 1k results if a model had experiments published on both 1k and 10k for data efficiency. References from left to right: (Sukhbaatar et al., 2015) - (Liu & Perez, 2017) - (Henaff et al., 2017) - (Seo et al., 2017) - Ours - (Xiong et al., 2016) - (Graves et al., 2016) - (Weston et al., 2015) - Ours - (Kumar et al., 2016)

| Support Size Model | Weak | | | | | | | Strong | | |
| | 1k | | | | | 10k | | 1k | | 10k |
| | N2N | GN2N | EntNet | QRN | UMN | DMN+ | DNC | MemNN | UMN | DMN |
|---|---|---|---|---|---|---|---|---|---|---|
| 1 | 0.0 | 0.0 | 0.7 | 0.0 | 0.0 | 0.0 | 0.0 | 0.0 | 0.0 | 0.0 |
| 2 | 8.3 | 8.1 | 56.4 | 0.5 | 65.6 | 0.3 | 0.4 | 0.0 | 0.7 | 1.8 |
| 3 | 40.3 | 38.8 | 69.7 | 1.2 | 67.1 | 1.1 | 1.8 | 0.0 | 2.4 | 4.8 |
| 4 | 2.8 | 0.4 | 1.4 | 0.7 | 0.0 | 0.0 | 0.0 | 0.0 | 0.0 | 0.0 |
| 5 | 13.1 | 1.0 | 4.6 | 1.2 | 3.4 | 0.5 | 0.8 | 2.0 | 1.1 | 0.7 |
| 6 | 7.6 | 8.4 | 30.0 | 1.2 | 0.2 | 0.0 | 0.0 | 0.0 | 0.0 | 0.0 |
| 7 | 17.3 | 17.8 | 22.3 | 9.4 | 22.0 | 2.4 | 0.6 | 15.0 | 11.3 | 3.1 |
| 8 | 10.0 | 12.5 | 19.2 | 3.7 | 10.3 | 0.0 | 0.3 | 9.0 | 7.6 | 3.5 |
| 9 | 13.2 | 10.7 | 31.5 | 0.0 | 0.1 | 0.0 | 0.2 | 0.0 | 0.0 | 0.0 |
| 10 | 15.1 | 16.5 | 15.6 | 0.0 | 0.1 | 0.0 | 0.2 | 2.0 | 0.3 | 2.5 |
| 11 | 0.9 | 0.0 | 8.0 | 0.0 | 0.0 | 0.0 | 0.0 | 0.0 | 0.0 | 0.1 |
| 12 | 0.2 | 0.0 | 0.8 | 0.0 | 0.0 | 0.0 | 0.0 | 0.0 | 0.0 | 0.0 |
| 13 | 0.4 | 0.0 | 9.0 | 0.3 | 2.1 | 0.0 | 0.1 | 0.0 | 0.1 | 0.2 |
| 14 | 1.7 | 1.2 | 62.9 | 3.8 | 19.7 | 0.2 | 0.4 | 1.0 | 0.1 | 0.0 |
| 15 | 0.0 | 0.0 | 57.8 | 0.0 | 0.0 | 0.0 | 0.0 | 0.0 | 0.0 | 0.0 |
| 16 | 1.3 | 0.1 | 53.2 | 53.4 | 55.2 | 45.3 | 55.1 | 0.0 | 0.0 | 0.6 |
| 17 | 51.0 | 41.7 | 46.4 | 51.8 | 39.2 | 4.2 | 12.0 | 35.0 | 49.3 | 40.6 |
| 18 | 11.1 | 9.2 | 8.8 | 8.8 | 4.4 | 2.1 | 0.8 | 5.0 | 0.5 | 4.7 |
| 19 | 82.8 | 88.5 | 90.4 | 90.7 | 91.8 | 0.0 | 3.9 | 64.0 | 46.7 | 65.5 |
| 20 | 0.0 | 0.0 | 2.6 | 0.3 | 0.0 | 0.0 | 0.0 | 0.0 | 0.0 | 0.0 |
| Mean | 13.9 | 12.7 | 29.6 | 11.3 | 19.1 | **2.8** | 3.8 | 6.7 | 6.0 | 6.4 |
| # > 5% | 11 | 10 | 15 | 5 | 8 | 1 | 2 | 4 | 4 | 2 |

Table 8: Individual task error rates (%) on the logical reasoning dataset.

| Size | 1k | | | | 10k | | | | 50 | | 20k | |
| Support | Weak | | Strong | | Weak | | Strong | | | | Weak | |
| Arity | 1 | 2 | 1 | 2 | 1 | 2 | 1 | 2 | 2 | 2 | 2 | 2 |
| # Invs / Model | 1 | 3 | 1 | 3 | 1 | 3 | 1 | 1 | 3 | 3 | DMN | IMA |
|---|---|---|---|---|---|---|---|---|---|---|---|---|
| Facts | 1.2 | 0.9 | 0.0 | 0.4 | 0.0 | 0.0 | 0.0 | 0.0 | 0.0 | 33.5 | 0.0 | 0.0 |
| Unification | 0.0 | 10.3 | 0.0 | 10.8 | 0.0 | 0.0 | 0.0 | 0.0 | 0.0 | 41.3 | 13.0 | 10.0 |
| 1 Step | 50.3 | 49.8 | 4.4 | 20.0 | 1.2 | 27.8 | 0.1 | 1.3 | 5.7 | 50.2 | 26.0 | 2.0 |
| 2 Steps | 47.5 | 50.0 | 5.7 | 35.0 | 37.2 | 47.8 | 0.0 | 29.7 | 28.7 | 49.9 | 33.0 | 5.0 |
| 3 Steps | 47.6 | 49.2 | 10.4 | 38.7 | 39.6 | 45.6 | 0.0 | 26.0 | 26.1 | 48.3 | 23.0 | 6.0 |
| AND | 31.3 | 37.4 | 10.7 | 16.4 | 29.8 | 29.0 | 0.2 | 0.4 | 1.2 | 50.0 | 20.0 | 5.0 |
| OR | 25.2 | 38.1 | 21.0 | 35.0 | 20.5 | 30.2 | 4.4 | 20.6 | 17.4 | 47.6 | 13.0 | 3.0 |
| Transitivity | | 50.0 | | 26.6 | | 39.6 | | 5.0 | 6.0 | 49.2 | 50.0 | 50.0 |
| 1 Step NBF | 46.4 | 38.7 | 3.8 | 28.8 | 1.1 | 21.6 | 0.1 | 1.1 | 8.0 | 47.6 | 21.0 | 2.0 |
| 2 Steps NBF | 48.5 | 48.9 | 7.7 | 39.6 | 30.4 | 48.2 | 0.1 | 33.4 | 28.7 | 50.3 | 15.0 | 4.0 |
| AND NBF | 51.0 | 50.1 | 43.1 | 48.6 | 29.4 | 44.2 | 0.1 | 1.3 | 40.1 | 49.5 | 16.0 | 8.0 |
| OR NBF | 51.4 | 48.4 | 50.8 | 47.3 | 47.6 | 47.8 | 21.3 | 27.6 | 30.5 | 47.3 | 25.0 | 14.0 |
| Mean | 36.4 | 39.3 | 14.3 | 28.9 | 21.5 | 31.8 | **2.4** | 12.2 | 16.0 | 47.1 | 21.2 | 9.1 |
| Std | 18.7 | 15.9 | 16.4 | 14.1 | 17.1 | 16.7 | 6.1 | 13.2 | 13.6 | 4.7 | 12.3 | 13.4 |
| # > 5% | 9 | 11 | 7 | 11 | 7 | 10 | 1 | 5 | 9 | 12 | 11 | 5 |

$$\frac{X \text{sandra went back to the } Y\text{:bathroom}}{\frac{\text{is } X\text{:sandra in the } Y\text{:bathroom}}{\text{yes}}}$$

Figure 7: bAbI task 6, yes or no questions. The invariant does not variablise the answer.

$$
\begin{aligned}
\textbf{\textit{X}}\text{:m} ( \textbf{\textit{Y}}\text{:e} ) &\vdash \textbf{\textit{X}}\text{:m} ( \textbf{\textit{Y}}\text{:e} ) \\
\textbf{\textit{X}}\text{:a} ( \textbf{\textit{Y}}\text{:w} , \textbf{\textit{Z}}\text{:e} ) &\vdash \textbf{\textit{X}}\text{:a} ( \textbf{\textit{Y}}\text{:w} , \textbf{\textit{Z}}\text{:e} ) \\
\textbf{\textit{X}}\text{:m} ( \mathtt{T} ) &\vdash \textbf{\textit{X}}\text{:m} ( \text{c} ) \\
\textbf{\textit{X}}\text{:x} ( \mathtt{A} ) \leftarrow \textit{not}\ \textbf{\textit{Z}}\text{:q} ( \mathtt{A} ) &\vdash \textbf{\textit{X}}\text{:x} ( \textbf{\textit{Y}}\text{:z} )
\end{aligned}
$$

Figure 8: Invariants learned on tasks 1, 2 and 11 with arity 1 and 2 from the logical reasoning dataset. Last invariant on task 11 lifts the example around the negation by failure, denoted as *not*, capturing its semantics.

|  |  |  |  |  |
|---|---|---|---|---|
| 3 **X** 7 **Y** head | | **X** 0 0 | 0 **Y** 0 | 6 4 **Y** |
| 7 4 **X** **Y** head | | 0 1 0 | 1 **X** 8 | 0 **X** 8 |
| **X** 3 1 **X** tail | | 0 0 **Y** | 0 2 0 | 0 0 7 |
| **X** **X** 5 **Y** duplicate | | head | centre | corner |

(a) Invariants with extra varialbes learned with UMLP.    (b) Mismatching invariants learned with UCNN.

Figure 9: Invariants learned that do not match the data generating distribution from UMLP and UCNN using $\le 1000$ examples to train. In these instances the unification still bind to the the correct symbols in order to predict the desired answer; quantitatively we get the same results. Variable default symbols are omitted for clarity.

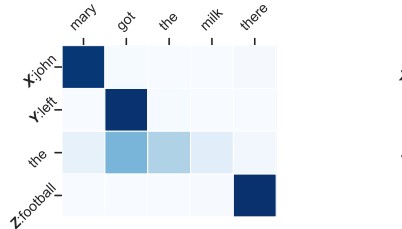 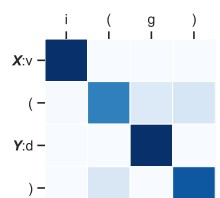 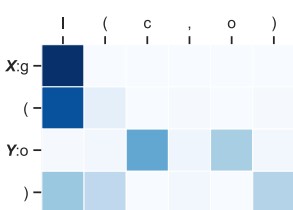

(a) bAbI task 2. When a variable is unused in the next iteration, e.g. **Z**:football, it unifies with random tokens often biased by position.

(b) Logical reasoning task 1. A one-to-one alignment is created between predicates and constants.

(c) Logical reasoning task 3. Arity 1 atom forced to bind with arity 2 creates a one-to-many mapping.

Figure 10: Further attention maps for equation 3, darker cells indicate higher attention values.

