# OpenReview forum: "Learning Invariants through Soft Unification"
_ICLR.cc/2020/Conference — Reject_

### Official Review · AnonReviewer3 · 2019-10-23
**Official Blind Review #3**

**Rating:** 1

**Review:**

This paper explores a very interesting idea: can a model learn what variables are and how to use them? Unfortunately, the paper doesn't seem quite ready: the model description was very hard to follow and it's not clear the approach has found a compelling use case.

I read the paper carefully three times, and try as I might, I simply can't get my head around the entire architecture. The modeling section jumps straight into a series of definitions, without trying to build intuition or provide a worked example. There is an example in Figure 2, but it isn't really explained and I didn't find it helpful. Unification seems to be implemented as a form of attention (or self-attention) where the model can control the degree to which a symbol acts as variable. But the relationship between soft unification and attention isn't really spelled out -- what's the same, what's different? Ultimately it's not clear to me what the model is attending over during soft unification.

There are various other aspects of the paper that aren't clear:
- strong vs. weak supervision
- comparison models DMN and IMA are not introduced at all, and include no references
- the logical reasoning experiment is not clearly described
- there is only a cursory conclusion

I am not sure the model has found a compelling use case. On bAbi with weak supervision, the model is worse than the comparison models. It only slightly beats out memory networks with strong supervision. For logical reasoning, it's not clear what it is compared against or if the comparison is fair. The clearest win over standard networks is on the simple synthetic experiments.

Finally, the authors mention the paper has a cognitive science motivation, in that "Humans learn what variables are and how to use then at a young age" or that "symbolic thought with variables is learned...", taking a strong "nurture" stance on the origin of variables. But variables could very well be innate and simply early emerging. Any discussion of the origin of variables in the mind requires more nuance.

I am excited about this research direction, and it could ultimately be a very nice contribution as the work matures. I don't think the paper is ready in its current form.


**Experience Assessment:**

I have published one or two papers in this area.

**Review Assessment: Checking Correctness Of Derivations And Theory:**

N/A

**Review Assessment: Checking Correctness Of Experiments:**

I carefully checked the experiments.

**Review Assessment: Thoroughness In Paper Reading:**

I read the paper at least twice and used my best judgement in assessing the paper.

---

> ### Author Response · Authors · 2019-11-10
> **Response to Review #3**
>
> Dear reviewer, thank you for your feedback. We are excited that you find the idea interesting and important. To answer your questions and comments:
>
> “But the relationship between soft unification and attention isn't really spelled out -- what's the same, what's different?” - The difference / similarity is mentioned multiple times in the paper, firstly in the introduction: “we consider unification a selection of the most appropriate value .., we can reframe it as a form of attention.” ; secondly in Section 2: “For example, … $\phi_V(X:s_d)$ would become a weighted sum of symbol embeddings as in conventional attention models”, and finally in Definition 3 where soft unification is defined as a dot product attention, equation 3. Hence, soft unification is implemented as a form of dot product attention.
>
> “Ultimately it's not clear to me what the model is attending over during soft unification.” - Following equation 3, soft unification attends over the symbols present in the example K. This K is another example from the dataset and could be a sequence, grid, a story or a logic program as setup in the datasets section and detailed for each architecture in Section 3.
>
> “strong vs. weak supervision” - We mention the strongly supervised experiments in Section 5: “in the strongly supervised cases, the negative log-likelihood for the context attentions are also added to the objective function.” In the memory networks literature surrounding the bAbI dataset, this refers to using the supporting facts. We do not supervise the soft unification mechanism in any of the experiments.
>
> “comparison models DMN and IMA are not introduced at all, and include no references” - We do not provide detailed previous work to reduce clutter, distinguish our work and adhere to space constraints. The reference for them are in the caption of Table 3: “and DMN, IMA by Cingillioglu & Russo (2019).” We ask the readers to refer to the cited paper for further details. Similarly N2N, GN2N, EntNet are not details and we ask the readers to refer to the citations.
>
> “the logical reasoning experiment is not clearly described” - We use an existing data generation procedure as mentioned, “using the procedure by Cingillioglu & Russo (2019).“ and only give details of the specific settings we used to generate the data such as the arity and size of the data. Similar to the bAbI dataset, for the logical reasoning dataset we ask readers to refer to the original papers that introduce the individual tasks.
>
> “there is only a cursory conclusion” - This is quite subjective as the conclusion of the paper clearly states a novel approach to incorporating variables to neural network architectures and learning invariants. We present the concrete output of this approach: the invariants learnt by analysing soft unification mechanism implemented as an attention.
>
> “I am not sure the model has found a compelling use case.” - This is interesting as the judgement seems to be made on the final accuracy based performance of the model in the bAbI and the logical reasoning dataset. The objective is “learning invariants” rather than to lower error rates further. We urge the readers to consider the qualitative novel output of our approach instead of a win or lose against other models in certain datasets. Our approach is flexible in the network architecture (UMLP, UCNN, UMN) as well as the tasks it can solve, in some cases  better or as good as other models.
>
> “But variables could very well be innate and simply early emerging.” - This is also very interesting, it may very well be. We followed the line of work cited in the introduction and the related work to establish our argument showing evidence such as pretend play etc. as to why the notion of a variable could be learned. The discussion about whether it could be innate is more appropriate in the field of developmental psychology that is outside the scope and focus of this work. We would be happy to incorporate references showing evidence for the innateness of variables in human reasoning.
>
> Please note that, Reviewer 4 supports the motivation of a cognitive background and Reviewer 1 points out the paper is well written, structured and clear. These seem to counter your two main concerns. We fail to find in the review any further scientific or technical grounds for disputing the validity or novelty of the work to hamper its publication. We believe we have addressed your questions and comments and highlighted the novelty and contribution this work brings.

---

### Official Review · AnonReviewer1 · 2019-10-23
**Official Blind Review #1**

**Rating:** 3

**Review:**

This paper presents a novel approach for learning invariants that can capture underlying patterns in the tasks through Unification Networks. This effectively allows the machine to learn the notion of `variable`, which is a symbol that can take on different values.

Pros:
The authors evaluated and presented empirical results on four common benchmark datasets, showing superiority over plain baseline without unification.
They further performed analysis on the learned invariants, and verified the sensibility.
The paper overall is well written and structured.

Cons:
Despite its superiority over plain baseline, the paper does not provide thorough comparison with other state-of-the-art methods on reasoning related tasks.

Some of the technical details regarding the choice of hyperparameters are missing. For example:
In section 6, what’s the rationale of setting $t$ differently for bAbI solely?
In Equation 5, how is the sparsity regularization parameter $\tau$ chosen optimally for a particular task? A bit more discussion on these choices would be helpful.

Overall, this paper presents a seemingly promising architecture capable of learning and using variables, with the caveat for lack of experiments and comparison with other state-of-the-art methods.


**Experience Assessment:**

I do not know much about this area.

**Review Assessment: Checking Correctness Of Derivations And Theory:**

N/A

**Review Assessment: Checking Correctness Of Experiments:**

I assessed the sensibility of the experiments.

**Review Assessment: Thoroughness In Paper Reading:**

I read the paper at least twice and used my best judgement in assessing the paper.

---

> ### Author Response · Authors · 2019-11-10
> **Response to Review #1**
>
> Dear reviewer, thank you for your feedback and we are glad you have found the paper well written and structured. To answer your questions and comments:
>
> “the paper does not provide thorough comparison with other state-of-the-art methods on reasoning related tasks” - If we are referring to the bAbI and the logical reasoning tasks as reasoning related tasks, we provide a detailed comparison of each task with all of the state-of-the-art models in Appendix D Table 7 and 8. We present similar (related to our memory network architecture) memory based architectures in the main body of the paper and the rest of the state-of-the-art models in the appendix due to space limitations.
>
> “In section 6, what’s the rationale of setting $t$ - threshold differently for bAbI solely?” - This question is answered at the beginning of Section 6, “The magnitude of this threshold seems to depend on the amount of regularisation, equation 5, and the number of training steps along with batch size all controlling how much $\psi$ is pushed towards 0.” Thus, after training is complete there will be a lower bound on $\psi$ depending on those aspects which is different for bAbI from the other datasets.
>
> “how is the sparsity regularization parameter $\tau$ chosen optimally for a particular task?” - It is not chosen optimally, we used 0.1 as a reasonable coefficient in recognising that $\tau$ is an L1 regularisation applied to $\psi$. We haven’t performed hyper-parameter tuning.
>
> “the caveat for lack of experiments and comparison with other state-of-the-art methods” - We disagree with this statement as we present 4 datasets, 3 different architectures, different experimental setups (strong vs weak, 1k vs 50 training examples), analysis of invariants, analysis of soft unification as well as comparison to existing state-of-the-architectures in Sections 4, 5 and 6 respectively with detailed results and further analysis in Appendix D.
>
> We hope we have answered your questions individually and highlighted the novelty, the results of the experimental setup and comparison to the state-of-the-art models presented in this work.

---

### Official Review · AnonReviewer4 · 2019-10-24
**Official Blind Review #4**

**Rating:** 3

**Review:**

The authors propose a neural network approach to variable unification and
reasoning by example as a way to mimic the human ability to identify invariant
patterns in examples and then apply them more generally in practice.
This general idea of identifying invariates and mapping new instances to
them is well motivated by the authors, citing work in philosophy of mind,
cognitive science, and developmental psychology.

The authors go on to propose MLP, CNN and Memory Network models
of unification for sequence, grid, and story reasoning tasks respectively.
Experiments on the sequence and grid datasets demonstrate the data efficiency
of this approach. MLP and CNN models with unification achieve near perfect
performance in fewer iterations (an order of magnitude fewer in the MLP case!)
 than their non-unification enabled counter parts.
Unification enabled models also demonstrate high performance in a reduced
training set setting (using only 50 training examples).
While this is encouraging, these are very simple toy tasks.

I also am in doubt as to whether the representation of these problems
causes some issues. In the sequence task, one question the models are
trying to solve is what symbol is the head or tail of the sequence.
Modeling variables over the sequence of symbols here is, in a sense, the
wrong object of study. The position of the symbols would need to be
represented, e.g.

a b c d
1 4 3 1

where I've represented positions as a-d, and the learned invariant about
head questions would be:

X:a b c d
Y:1 4 3 1

As is, by mapping symbols and not positions to variables, one cannot,
at the variable level distinguish between the two 1s in the sequence above.
My guess is that in practice the bi-GRU model that produces embedding
features of the symbols in sequence is implicitly representing head/tail
positioning.

Similar arguments could be made about the grid example.

I don't find the experiments/analysis on the bAbI dataset very convincing.
For instance, in the example given in Figure 4b (reproduced below)
is shown as an example of
temporal reasoning, where a symbol Z is mapped to the
word morning (a symbol distinguishing a time), and the question asked is
where was Bill before school.
If logical reasoning is being used to solve this question, surely the
symbol 'before' must also be represented as a variable. Its possible that
the model is instead learning a trick about mutual exclusivity, i.e. that
Y:school is the only location symbol not mentioned in question but this
could fail as a general strategy.


this Z:morning X:bill went to the Y:school
yesterday X:bill journeyed to the A:park
where was X:bill before the Y:school
A:park

Figure 4b

It would make for a much more interesting paper if the authors took
examples such as these and formed counter-factuals to probe the way
the models are answering the questions. E.g., transforming the question
in 4b to "where was X:bill today" or "where was X:bill after school."

Because the authors use soft unification, interpretability is difficult
to assess. Interpretability is crucial here because to claim that unification and reasoning by logical induction
is being used to solve tasks, it becomes important to show how the neural networks
make their decisions. Given the instances of extra variables and one to many
mappings on the bAbI dataset it seems very likely that the models are not
solving many tasks
using unification as it would be possible to learn to use the symbols directly
to learn to answer. As such, I think these issues are not addressed in the
paper sufficiently to warrant acceptance.


Minor Notes

- In definition 1, the definition of Variable is a little confusing because there are two different senses of the word in use. I understand them to be (1) Variable (X) in the logical template that is intended to be learned and used in problem solving, and
(2) variable (x) in the neural network model that is a soft asignment of
the Variable to a default symbol s. It would be nice if this distinction could
be noted or made clearer.

- In the definition 2, in the phrase "is the invariant example such as a tokenized story" it might be worth stating that the tokens are the symbols in S.


- My understanding is that each unique symbol in the invariate is a potential
variable. Does this mean there are no co-referent symbols in the invariate?
Would be helpful to state whether babi contains co-referent expressions
and how these might affect the model.


- It might be interesting to see how model architecture affects variable
learning. For example, does a CNN result in more sensible variable
assignments  than the mlp on a flattened representation of the grid problem?


- What is the strongly supervised case? These are token level annotations I think (at least for babi) but it might be good to specify in more detail what
they  are.

- The figure and explanation of the UMN are not very clear. From the figure
is does not seem that the variables interact with the memory at all. More
space could be devoted to this section.

Possibly Relevant Related Work

Brenden Lake. Compositional generalization through metasequence-to-sequence learning. NeurIPS 2019.







**Experience Assessment:**

I have read many papers in this area.

**Review Assessment: Checking Correctness Of Derivations And Theory:**

I carefully checked the derivations and theory.

**Review Assessment: Checking Correctness Of Experiments:**

I carefully checked the experiments.

**Review Assessment: Thoroughness In Paper Reading:**

I read the paper thoroughly.

---

> ### Author Response · Authors · 2019-11-10
> **Response to Review #4**
>
> Dear reviewer, thank you for your detailed constructive feedback. We are happy to answer your comments:
>
> “these are very simple toy tasks” - We must first understand how these architectures work and analyse them in a controlled environment in order to mitigate the black box effect they create. In order to analyse the invariants, it is crucial to work in a fixed setting where the data generating distribution is known for comparison. This learning task has never been attempted before and the paper demonstrates the validity of the idea and its effectiveness, enabling work towards addressing more complex tasks.
>
> “Modelling variables over the sequence of symbols here is, in a sense, the wrong object of study” - Our approach is indifferent to the underlying structure of the task, demonstrated by the different datasets. There is no right or wrong task with respect to learning variables since the model does not make assumptions about the data. We present 3 different structures still using the same definitions from Section 2. The soft unification function “g” potentially learns different features with different structures, as you have guessed in the next comment below.
>
> “My guess is that in practice the bi-GRU model that produces embedding features of the symbols in sequence is implicitly representing head/tail positioning.” - Your guess is correct! We state this in Section 2 as the unifying properties $\phi_U$ that can be learned. In the example you’ve provided, 1 4 3 1, the unifying features of 1s will be different due to the bi-GRU. This also gives great capacity to the network and the ability to unify the head of a sequence with the tail of another sequence (Appendix D).
>
> “If logical reasoning is being used to solve this question, surely the symbol 'before' must also be represented as a variable” - We must be careful in projecting our understanding of natural language and logical reasoning to dictate what should and shouldn’t be a variable. If this was an alien language with unknown symbols, we wouldn’t be able to say a symbol should be a variable nor assume a certain logical reasoning is involved. Hence, only variations in the data can tell, as is the case in our approach, whether a symbol should be a variable. Since the model can optimise to use 1 variable where we might expect 2, Figure 5b, it might not follow the data generating distribution exactly but still solve the task by exploiting these commonalities. We discuss this in “Interpretability versus Ability”, Section 6.
>
> “.. formed counter-factuals to probe the way the models are answering the questions” - This is an interesting point we also make in our relevant work in Section 7. Our objective of learning invariants to some extent uses counter-factuals as unification changes the facts of a story. Furthermore, your question is not a counter-factual as it can be answered with “unknown” looking at the story facts. The question “where would X:bill have been before Y:school should he have gone to the garden yesterday” is a counter-factual as it yields an answer of garden against the facts presented in the story.
>
> “Interpretability is crucial here because to claim that unification and reasoning by logical induction is being used to solve tasks ...” - We don’t claim this is logical unification, reasoning or induction. In fact, we refrain from using those terms “since neither the invariant structure needs to be rule-like nor the variables carry logical semantics”. The reason is because the “g” and the “f” are learned end-to-end and could learn elements of logical unification, reasoning or not; hence, it is inappropriate to assume or claim that it is any sort of logical induction.
>
> “Does this mean there are no co-referent symbols in the invariate?” Yes, each symbol is considered unique and a potential variable independently. In bAbI co-reference task,11, “He” etc. are unique symbols and treated equally. Detailed results, including task 11, are in Appendix D Table 6.
>
> “does a CNN result in more sensible variable assignments  than the mlp on a flattened representation of the grid problem?” - It is a good question. It depends on what we mean by sensible. If we refer to them following the data generating distribution then the answer is similar to asking whether an MLP or a CNN solves the task better. This is because the variables are learned with respect to an upstream “f” and that network provides the gradients for which symbols should be variables. In either case, we observe occasional “insensible” variables (Appendix D Figure 9) in which the invariants can still solve the task.
>
> “What is the strongly supervised case?” - These experiments use the supporting facts provided in the bAbI dataset as done in literature around memory networks. This is mentioned in Section 5: “and, in the strongly supervised cases, the negative log-likelihood for the context attentions are also added to the objective function.” We do not supervise soft unification nor label correct tokens.

---

### Decision · Program_Chairs · 2019-12-19

**Decision:**

Reject

**Comment:**

The main concern raised by reviewers is the limited experiments, which are on simple tasks and missing some baselines to state-of-the-art methods. While the overall approach is interesting, the reviewers found the empirical evidence to be fairly unconvincing.